# A Systematic Analysis of Recent Technology Trends of Microfluidic Medical Devices in the United States

**DOI:** 10.3390/mi14071293

**Published:** 2023-06-24

**Authors:** Rucha Natu, Luke Herbertson, Grazziela Sena, Kate Strachan, Suvajyoti Guha

**Affiliations:** Division of Applied Mechanics, Office of Science and Engineering Laboratories, Center for Devices and Radiological Health, Silver Spring, MD 20993, USA

**Keywords:** medical devices, microfluidics, low flow systems, bench testing, adverse events, modes of failure

## Abstract

In recent years, the U.S. Food and Drug Administration (FDA) has seen an increase in microfluidic medical device submissions, likely stemming from recent advancements in microfluidic technologies. This recent trend has only been enhanced during the COVID-19 pandemic, as microfluidic-based test kits have been used for diagnosis. To better understand the implications of this emerging technology, device submissions to the FDA from 2015 to 2021 containing microfluidic technologies have been systematically reviewed to identify trends in microfluidic medical applications, performance tests, standards used, fabrication techniques, materials, and flow systems. More than 80% of devices with microfluidic platforms were found to be diagnostic in nature, with lateral flow systems accounting for about 35% of all identified microfluidic devices. A targeted analysis of over 40,000 adverse event reports linked to microfluidic technologies revealed that flow, operation, and data output related failures are the most common failure modes for these device types. Lastly, this paper highlights key considerations for developing new protocols for various microfluidic applications that use certain analytes (e.g., blood, urine, nasal-pharyngeal swab), materials, flow, and detection mechanisms. We anticipate that these considerations would help facilitate innovation in microfluidic-based medical devices.

## 1. Introduction

Microfluidics involves the manipulation and control of fluid flows in micron and submicron sized channels. Microfluidic-based devices offer potential advantages over traditional macroscale techniques, such as small sample volumes, low reagent consumption, reduced analysis times, improved interaction efficiency due to larger surface area-to-volume ratios, simplified user-friendly procedures, and portability [1]. Microfluidic technologies could transform biomedical processes, such as sample handling methods, sample preparation, diagnosis, and analytical techniques used in laboratories. Lab on a chip applications enable complex processes—from sample pretreatment to sample manipulation and analysis—to be conducted in an automated, plug-and-play fashion on a single chip without user intervention. Due to these advantages, in the past few decades, the field of microfluidics has had a positive impact on different industries, such as fuel cells [2], bio-sensing [3], environmental monitoring [4], micro-reactors [5], gas chromatography [6], and biomedical applications [7]. Biomedical areas in which microfluidics may have the most immediate impact include point-of-care diagnostics, sepsis detection, cancer diagnosis, bio-sensing, pharmaceutical analysis, pharmacological testing, and detection of cardiovascular disorders [8].

In recent years, the Center for Devices and Radiological Health (CDRH) within the U.S. Food and Drug Administration (FDA) has received an increasing number of medical device submissions that incorporate microfluidic technologies [9]. A medical device is defined as any “instrument, apparatus, implement, machine, implant, or other similar or related article, including a component part, or accessory which is intended for use in the diagnosis of disease or other conditions, or in the cure, mitigation, treatment, or prevention of disease… and does not achieve its primary intended purpose through chemical action within or on the body of man or other animals and which is not dependent upon being metabolized for the achievement of its primary intended purpose” [10]. A growing number of medical devices use novel microfluidic technologies that can augment or even replace conventional macroscale systems. To understand and evolve with technology trends, CDRH has become actively involved in regulatory science research of microfluidic-based medical devices to help facilitate the translation of laboratory-based systems to commercial products. Since microfluidics is still an emerging technology in the biomedical field, there are currently no FDA-recognized standards or FDA guidance documents developed specifically to aid in the evaluation of microfluidic-based medical devices [9]. Traditional approaches or existing standards used for macro-scale devices may not always be appropriate for assessing microfluidic technologies due to scalability [11].

Many published literature reviews have focused on specific microfluidic techniques and their applications in the biomedical field. A recent review by Niculescu et al. summarized different biomedical applications [12], fabrication techniques, and materials used for biomedical devices. Mahhengam et al. have discussed potential applications of microfluidics in cancer diagnosis and treatment [13], while Sachdeva et al. have described the commercial point-of-care device space [14]. Zhang et al. have focused on microfluidic methods employed for sepsis diagnosis [15]. Azizipour et al. have provided an overview of the development of biochips from lab on a chip to organ on a chip applications [16], while Feng et al. [17] focused on droplet microfluidics, which constitutes one of the largest application areas in the microfluidic-based medical device space. Inertial microfluidics and magnetic separation are topics that have been discussed in other recent technology specific reviews as well [18,19]. Most of these reviews present academic research-focused findings, rather than commercial use applications. Only a few articles describe the commercial landscape of microfluidic medical devices, the current challenges, trends in biomedical applications, and ways to overcome barriers to commercialization [20,21]. To date, few, if any, articles discuss the common modes of failure in microfluidic medical devices or comprehensively analyze test methods to inform future microfluidic-focused standards. Thus, the objectives of this article are to (i) analyze submissions that have either been cleared, approved, or authorized by the FDA to identify the commonly occurring applications of microfluidic-based medical devices, and (ii) analyze existing standards used and the common modes of failure to help identify the areas of need for new test methods and protocols that may aid in the evaluation of microfluidic medical devices. 

## 2. Materials and Methods

To determine recent microfluidic technology trends in the biomedical field and to effectively identify medical devices containing microfluidic components, a systematic review of medical device submissions to the FDA was performed. 

### 2.1. Identification and Inclusion of Submissions

A keyword-based search was performed (Figure 1) to retrospectively collect data from device submissions to CDRH. This datamining effort generated a list of more than 10,000 regulatory submissions containing the selected keywords “microfluidics”, “lab on a chip”, “capillary flow”, and “microchannel” indicating microfluidics components; however, not every identified submission incorporated microfluidic technology. For example, if a submission simply referenced a journal article pertaining to the keyword “microfluidics” without incorporating any microfluidic technology for the device operation, this file would not be considered pertinent to the study. To remove these files from the analysis, additional filtering criteria were needed (Figure 1). The file types considered for this analysis included premarket notifications (510(k)), De Novo classifications, pre-market approval (PMA) applications, and emergency use authorizations (EUA). Table 1 provides brief definitions of common regulatory pathways, processes, and classifications for medical devices in the United States. Using the inclusion criterion listed in Figure 1, more than 200 medical devices were analyzed. Class I devices, which constitute the lowest risk medical devices and only require general controls, such as for adulteration, misbranding, registration and listing, and good manufacturing practices [22], were excluded from this datamining effort. Investigational device exemptions (IDEs), pre-EUAs, and pre-submissions were excluded from the analysis since these interactions are not required to be made public. Only publicly available information for the included devices (e.g., information derived from publicly available 510(k) summaries and decision summaries) is discussed in this paper.

Due to a lack of consistency in microfluidics nomenclature within the medical device community, it was anticipated that some medical device manufacturers may not use the selected keywords in their regulatory submissions. Hence, to ensure that such devices were not overlooked, a separate Google Scholar search was performed to identify other commercially available microfluidics-based medical devices. Note, the keywords used for the Google Scholar search were different than those used in datamining the regulatory submissions (Figure 1), to improve the relevance of the search results. Two recently published books about microfluidics [23,24] were also reviewed to obtain pertinent information. Further, recent review articles that reported either commercial medical devices employing microfluidics technology or research performed at academic institutions that was subsequently adopted for commercialization [14,25] were included. After exploring these different ways for identifying potential legally marketed microfluidic-based medical devices, the publicly available FDA 510(k) device database was used to extract information about the regulatory submissions, along with FDA’s internal repository of medical device submissions. A microfluidics expert at FDA verified that all medical devices included in this datamining effort contained some aspect of microfluidic technology. In total, more than 300 devices were included in this comprehensive analysis of microfluidic-based medical devices.

**Figure 1 micromachines-14-01293-f001:**
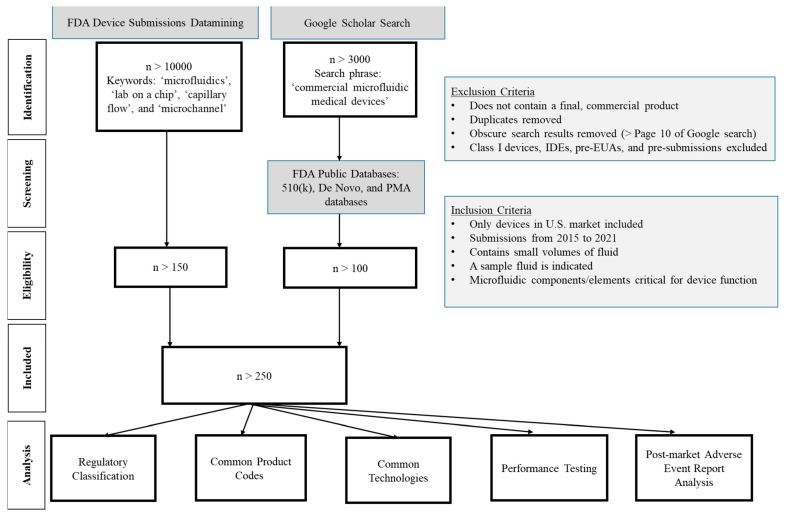
PRISMA flow diagram of the systematic analysis to identify medical devices that incorporate microfluidic technologies. The FDA public databases can be accessed using the 510(k) database [26], De Novo database [27] and PMA database [28] through www.fda.gov (accessed on 19 October 2022).

### 2.2. Data Analysis

After compiling a comprehensive dataset of medical device submissions containing microfluidics technology, spanning the timeframe from January 2015 to December 2021, an in-depth data analysis was performed in Microsoft Excel using pivot tables (Microsoft 365) to elucidate microfluidics trends in the biomedical space. Information about regulatory classification, product codes, application, device characteristics, performance testing, and failure modes was collected for all the microfluidics-based medical devices. 

#### 2.2.1. Regulatory Classification

Most of the key types of regulatory submissions for medical devices in the U.S. are described in Table 1. The regulatory classification or submission type can provide some indication for the level of risk or complexity associated with the device under investigation, as well as the current stage within the pathway to commercialization in which the device falls [29,30]. For instance, a PMA device is likely to have a long history of interactions with the regulatory agency to bring a high-risk, complex medical device to market.

**Table 1 micromachines-14-01293-t001:** Different regulatory submission types for microfluidic-based medical devices in the U.S. [31].

Regulatory Submission Types	Description
De Novo Classification Request	The De Novo classification is a pathway to classify novel medical devices for which general controls (e.g., registration, recalls, adverse event reporting) and special controls, such as performance standards and special labeling requirements, can provide reasonable assurance of safety and effectiveness for the intended use, but for which there is no legally marketed predicate device.
Emergency Use Authorization (EUA) and Pre-EUAs (PEUA)	FDA may authorize unapproved medical devices or unapproved uses of approved medical devices to diagnose, treat, or prevent serious or life-threatening diseases or conditions during a public health emergency, such as the global COVID-19 pandemic. A PEUA can precede a future or current EUA submission.
Investigational Device Exemption (IDE)	An IDE application allows for an investigational device to be used in a clinical study to collect safety and effectiveness data, which would subsequently enable a manufacturer to submit a device for clearance or approval. Preclinical test data are required to establish some level of safety for the IDE.
Premarket Approval (PMA)	The PMA pathway is a regulatory review process to evaluate the safety and effectiveness of primarily Class III medical devices (i.e., devices that support or sustain human life, are of substantial importance in preventing impairment of human health, or which present a potential, unreasonable risk of illness or injury).
Premarket Notification (510(k))	A 510(k) is a premarket submission to demonstrate that the device to be marketed is as safe and effective, or substantially equivalent, to a legally marketed device. Submitters must compare their device to one or more similar legally marketed devices to demonstrate substantial equivalence.

#### 2.2.2. Product Code

This three-letter descriptor indicates the category of a medical device often unique for specific medical device applications, indications for use, or technologies. More information about product codes can be found in the product code database [32]. The FDA provides a database of product codes so that the device manufacturers can determine what information is necessary for assessing the device type. Product codes were used to classify devices based on similar technological characteristics and for analyzing adverse events.

#### 2.2.3. Application

The microfluidic-based medical devices in this dataset have been categorized according to their application. By classifying based on application, we can identify common and unique ways that microfluidics are used in the biomedical field. The most prominent applications of microfluidic medical devices, to date, have been identified through this datamining effort.

#### 2.2.4. Device Characteristics

Microfluidic device characteristics were analyzed in terms of material properties, fabrication methods, technologies, fluid properties of the test sample, and flow characteristics. The material properties of the medical device are dictated by whether it is disposable or reusable, the fluid sample being analyzed, the complexity of the design, and the operating conditions under which the device is used. The microfluidic technology employed by the device enables unique aspiration, flow, sensing, and detection capabilities that may require standardized bench test methods. This datamining effort involved determining flow-related trends, including infusion and perfusion mechanisms and the need for microscale fluid contacting components. 

#### 2.2.5. Performance Testing

These specific types of test methods that have been used by companies to evaluate their microfluidic medical devices for safety and efficacy are typically identified in public device databases. These preclinical tests were analyzed to determine if specific testing is needed to fully assess the microfluidic aspects of medical devices. 

### 2.3. Failure Modes

A failure mode analysis was performed to determine if specific issues or complications are associated with microfluidic-based medical devices. The information about adverse events was collected from FDA’s publicly available Manufacturer and User Facility Device Experience (MAUDE) database [33] and sorted using Power BI (Microsoft) software. First, the adverse events reported for the most common applications were collected for the time period from January 2015 through September 2021. Based on their frequent occurrence amongst the microfluidic device submissions we had identified, the product codes JPA, OOI, NBW, OCC, GKZ, LZG, PAM, PEN, PEO, QFG, CGA, DKZ, FKJ, GJS, GTY, JSM, LCX, MEB, MQK, and PCH were used for this analysis. By subtracting the adverse events from the manufacturers that are known to use microfluidics, we were able to obtain and analyze the number of adverse events for devices that use traditional technologies that do not incorporate microfluidics. A second analysis was performed using the submission numbers of the microfluidic devices that shared the same product codes over the same time period. PMAs were excluded from this analysis due to a low number of corresponding adverse events. Categorizing the adverse events allowed us to determine if certain failure modes may be associated with microfluidics compared to other technologies. A separate analysis of product codes QKP, QJR, and QKO was also performed for COVID-19 EUA submissions that used microfluidic technologies.

## 3. Results

### 3.1. Trends in Regulatory Submissions

Overall, increasing trends in the number of microfluidic-based medical device publications and regulatory submissions have been observed since 2000, as shown in Figure 2a. While there have been hundreds of microfluidics-related journal articles published each year for the last two decades, the number of medical device submissions in this field have been orders of magnitude smaller. Yet, overall, the number of publications and medical device submissions have greatly increased in recent years, with 3-year moving averages of 27% and 96%, respectively, from 2015 to 2019. A steeper increase in regulatory submissions has emerged more recently, particularly in 2020–2021 as the time from feasibility to translation has been shortened with COVID-19 related EUAs that use microfluidics. As a result, the 2-year moving average for submissions to FDA for the years 2020 and 2021 has increased by 132%. 

As evident from Figure 2b, 510(k) devices and pre-submissions make up most of the microfluidics-based medical device submissions. Pre-submissions are a mechanism that allows manufacturers to engage early and often with the FDA to obtain feedback on the device evaluation strategy and test plan. For most of these devices, the comparator system used were also microfluidic based. More than 80% of the submissions in this analysis were for in vitro diagnostic devices. More than 95% of the EUAs shown here were submitted during 2020 and 2021 for COVID-19 related applications. Note that to protect sensitive, non-public information, information from pre-submissions, pre-EUAs, and IDEs were excluded (refer to Figure 1) from all further analysis. 

### 3.2. Microfluidic Device Applications

Table 2 lists the most common applications in descending order along with some of their important attributes. It is not surprising to see that the most commonly occurring applications were also in the COVID-19 diagnostic space. Before the COVID-19 pandemic, most regulatory submissions involving microfluidics were categorized as either 510(k) devices or pre-submissions. Hematology analyzers, blood filters, micro-dosing, and reproduction assist devices represent other prominent applications.

### 3.3. Microfluidics Device Characteristics

Figure 3a shows a plot of the percentages of devices which use common materials. It should be noted that about half of the submissions analyzed did not provide detailed information about the fabrication materials used in the device. Commonly used materials in microfluidic devices are highlighted in a recent review [12]. For the submissions that did include information about device materials, about 45% of the microfluidic systems were fabricated using plastic substrates. Commonly used plastics include acrylic, medical-grade polypropylene (PP), polymethyl methacrylate (PMMA), polycarbonate (PC), and polystyrene (PS). About 16% of microfluidic medical devices were made using glass, and polydimethylsiloxane (PDMS) was rarely used (~1%). About 45% of the devices used synthetic membranes for microchannels or as absorption pads. Only about 10% of devices used PTFE tubing in the flow pathway. Subcomponents were often made up of various materials. For example, filters were made up of PS or silicon nanopore membrane, conjugate pads were made with polymer fibers, and bladders and reservoirs were elastomeric. Test cards commonly used synthetic glass fiber or nitrocellulose membranes that were laminated to form microchannels. Printed circuit boards or silicon-based etched chips were also used as device substrates for some applications. About 15% of devices used gold or platinum electrodes in applications for electrical sensing or digital droplet manipulation, and some microfluidic components were coated with hydrophobic solutions or functionalized for detection of antibodies or chemical manipulation. Injection molding was found to be the most common fabrication technique for microfluidic-based technologies.

Figure 3b shows the different types of flow mechanisms used for microfluidic devices. The simplicity of capillary flow by the action of spontaneous wicking of the analyte in microfluidic spaces makes it the most popular (50%), and this flow mechanism was found to be widely used in lateral flow assays (LFAs) (Table 2). Most other mechanisms were typically part of complex systems that provide quantitative analysis (e.g., pneumatic flow that may control the movement or mixing of the analyte and reagents, if applicable in confined microfluidic channels) or were directly used for delivering therapeutic doses to patients (e.g., positive displacement pumps with syringes used in insulin pumps). In some cases, flow mechanisms also appeared in conjunction with other mechanisms (e.g., capillary flow can be used for introduction of an analyte or reagent followed by further processing of the sample inside an analyzer using pumps). As part of such systems, rotating valves, piston pumps, and plunger syringe barrels are also used for flow manipulation. Some device manufacturers did not go into specific details about the pumps used and they were all aggregated and categorized under pump (Figure 3b). Pneumatic flow using air or vacuum based transport with the help of inflatable bladders, flexible films, or membranes and air-filled microchannels was determined to be a prominent technique when the movement of the sample and reagents are restricted to the cartridge. Centrifugal discs that use centrifugal force from the rotation of disc to move the fluid into the microfluidic channel were used rarely (6%). Other techniques like electro-osmotic flow, droplet manipulation using electro-wetting and siphoning were used far less frequently and collectively constituted only 6% of the various flow mechanisms (Figure 3b). 

Technologies used in the devices are very diverse due to the broad range of applications (Table 2). Lateral flow assays form about 35% of the microfluidic devices seen at FDA. This number has increased especially during the COVID-19 pandemic. These assays are used in a large number in point of care and home use settings. Mechanical systems mostly relied on viscous forces or magnetic forces for particle manipulation. Micro-electromechanical devices were found to use electrical or electro-kinetic forces for sample manipulation. Different technologies were used for sample input, reagent flow, as well as for mixing. Sample input for a majority of the devices occurred by capillary action through glass tubes, microneedles, or absorption pads. About 62% of the devices used blood or blood derived fluids (human serum, plasma, blood cells) as input fluid. Other commonly used fluids include nasopharyngeal swab solutions, reagents, saliva, and urine. 

Both qualitative and quantitative detection/sensing technologies were used for obtaining output from devices (Table 2). Electrochemical sensing, visual detection and fluorescence detection were most common and were used by about 30%, 21%, and 16% devices, respectively. Visual detection included image analysis for detection of certain phenomena, like clotting, and particle sorting or chromatography techniques, like change in color/line or change in turbidity due to agglutination of particles. Electric current or resistance was used to detect presence of an agent of interest for 11% of the devices. 

### 3.4. Bench Testing

Performance testing is a critical aspect of device evaluation and is required in almost all devices (except low-risk devices). It can include non-clinical bench testing, as well as clinical tests [34,35]. There are no specific FDA-recognized standards for microfluidics, therefore manufacturers have had to modify existing standards or develop their own test methodologies. Traditional test methods, such as those used to assess chemical, thermal stability, mechanical, electrical, and software issues were unlikely to be impacted by the use of microfluidic technology. On the other hand, for shelf-life testing that assesses device functionality over the entire life cycle of devices, several manufacturers conducted component-level testing with electrodes, pumps, valves and sensors, media, filter paper, or membranes as applicable before testing on the entire device. Although not intended for small-scale devices, use of the Clinical and Laboratory Standards Institute (CLSI) guideline EP25-A [36] was common.

Flow testing was common for devices using positive displacement pumps (e.g., for micro-dose delivery of insulin for diabetes management, Table 2). Additional tests were also performed to determine dose-delivery accuracy under various worst-case conditions, such as temperature and pressure. The testing helped address any potential flow-related issues that may have emerged as a consequence of miniaturization. Relatively complicated devices in other application areas (Table 2) also required testing for device priming, pressure sensing, proper operation of the fluid circuit with the pump and tubing, as well as for analyzer-chip integration testing. 

Usability of a device often intersected with flow testing. In general, usability testing is carried out to ensure that the medical device is safe and effective for the intended user, uses and use-environment [37]. Some specific examples for various application types are provided in Table 2. For example, for addressing issues around sample introduction, several manufacturers characterized the input to the device, such as sample volume adequacy, sample handling quality of the sample (e.g., test for contamination, sample viscosity, proper aspiration) and nature of sample (bubble free, homogeneous). The reproducibility testing for the aspiration process at the inlet—for the case of manually applied or pipetted sample or sample absorbed by capillary action—was also conducted. In reusable lab-based diagnostic devices, usability assessments included a cleaning step for the sensors after every use to reduce the cross-contamination of samples. This type of additional measure is likely to be beneficial especially when using microfluidic technology since the increased surface area in microchannels may exacerbate biofouling and cross-contamination. 

Risk management [38] and failure mode analysis performed by manufacturers (Table 2) helped address the common modes of failure [39] seen with microfluidics. Leakage testing, connector strength testing, occlusion testing, and bubble detection evaluations were carried out by manufacturers in some cases. Some manufacturers used (ISO) 28620:2010 [40] and ISO 594-2 [41] for leakage testing. Some companies used International Electrotechnical Commission (IEC) 62366 [37] for testing the electrical systems for the devices. Custom flex studies [42] for fluid testing in devices that are often required as part of hazard analysis were also developed by manufacturers. Most manufacturers included the impact of transport, sample volumes, usability errors during loading, and temperature range of operation (Table 2) to address common issues pertaining to handling, transport, and temperature that are likely to be worse in microfluidic devices. 

Analytical studies using clinical samples are important for in vitro diagnostic devices [43]. We are not aware of any analytical studies that are uniquely required for microfluidics as these studies are typically technology independent. Hence, manufacturers leverage test methods used for legacy technologies. For example, linearity, limit of detection, precision, reproducibility, interference, sample stability, and storage testing is often performed using clinical and laboratory standard methods, specifically CLSI EP6-A [44], CLSI EP17-A2 [45], CLSI EP5-A3 [46], CLSI EP15-A3 [47], CLSI EP7-A2 [48], and CLSI EP25-A [36], respectively. 

### 3.5. Failure Mode Analysis

The adverse event query with microfluidic devices generated over 40,000 reports, while the second query with product codes generated about 1,180,000 reports specifically for non-microfluidic devices. Since the actual number of occurrences for the latter is about 30x higher, the data have been normalized and expressed in percentages for relative comparisons in Figure 4. Given the reasons behind failures are often reported in multiple ways, the failure modes were first consolidated and summarized in Table 3. These classifications were strictly made by the authors and do not represent any formal classifications of MDRs, or do not correlate with specific recalls, or recall types for the devices [49] discussed in this article. Operational failures (Figure 4) were high in both non-microfluidic as well as microfluidic. A higher percentage of flow-related failures (14% compared to 6%) were seen in the case of microfluidic devices. Analysis of COVID diagnostic EUAs for the time period of 2020–2021 generated approximately 14,000 reports and revealed that 89% of the failures reported are related to the data output or readout and 8% fell in the “Other” category (data not plotted). Within the data output category, false positive results were the most prominent failure mode, followed by incorrect, inadequate, or imprecise results, and false negative results. 

## 4. Discussion

Figure 2a shows that the number of microfluidic device publications each year is orders of magnitude greater than the number of medical device submissions to the FDA. An important reason for the large disparity in the number of research publications to commercial medical devices is the difficulties encountered when translating any new technology from research labs to medical devices for commercial use. Specifically, for microfluidics, several studies [20,50,51,52] have identified challenges that may explain the disparity in number of publications with submissions to the FDA (Figure 2). These challenges include (1) difficulty in translating lab-based fabrication to large scale commercial fabrication; (2) lack of modularity; (3) absence of applicable standards; (4) issues in integration with external macro systems; and (5) failure modes pertaining to issues such as microflow, bubble formation, clogging, and leakage. To understand the current areas of need, we start with the application specific challenges. 

### 4.1. Application Specific Challenges, Failure Modes, and Areas of Need 

Most of the applications using microfluidics technology pertain to microbiology, followed by applications in diabetes, blood coagulation, pregnancy test kits, and ion detection. As of December 2021, CDRH received more than 3700 original EUA requests and 2600 pre-EUAs, which included 290 molecular and nearly 90 antibody-based in vitro diagnostic tests pertaining to COVID-19 [53]. Not surprisingly, the most common applications for microfluidics are also related to COVID-19 (Table 2). The applications in Table 2 are organized in descending order of the number of device submissions received and analyzed. The type of application and fluid type often dictates whether the device can be operated by the patient at home, or if a trained technician or clinician is required to properly use the device. 

In general, many microbiology applications and pregnancy test kits use LFAs. LFAs are qualitative, paper-based assays used for obtaining rapid results, in which the sample is absorbed on a sample pad and transfers to the reaction site through paper-based microchannels driven by capillary forces. In these cases, the biochemical interaction, antigen–antibody binding, or probe DNA-target DNA binding occurs within the microchannel. The LFA devices usually consist of a sample pad, conjugate pad for labeled bio-recognition elements, reaction membrane, and waste pad [54]. Detailed information on the make-up of these assays and the working principle can be found in the review article by Koczula et al. [55]. The LFA test results are interpreted visually after a pre-specified amount of time by observing a change in color or the appearance of a test line. While LFAs have been used for decades, they are becoming increasingly sophisticated with the introduction of smaller sample volumes and patterning to create multiple flow channels [56]. Their simplistic design, low consumable costs, and excellent agreement with gold standard methods [57] are likely to continue to make capillary flow-based LFAs a popular technology of choice by device developers. 

The design simplicity of many micro-scale systems may also explain why the number of electrical, software, and structural failures are relatively small for microfluidics-based medical devices (Figure 4). Several of the most common applications typically require sample volumes of <10 µL and short analysis times <10 min. While small sample volumes, quick analysis times, and rapid scalability are some of the major benefits for using LFAs, these same factors can also make the devices more prone to errors involving reproducibility, sampling, and usability. Systems with smaller flow pathways may also be more susceptible to mechanical shock, and fluctuations in temperature and humidity, during transport or storage, which may adversely affect their performance. Therefore, it is important to find an optimal balance between miniaturization and designing an efficient and reproducible device. In addition, it may be advantageous to have performance testing protocols specifically developed for addressing these types of challenges around LFAs and other microfluidic applications [58] instead of continuing to leverage existing, more general standards in all cases (Section 3.4). Given that pregnancy test systems are LFA-based, some of the challenges associated with these devices are likely to be similar to those of COVID-19 diagnostic tests.

Due to limited test sensitivity and quantitation of LFAs, other operating techniques, such as fluorescence, electrochemical or electrical signal-based detection, are used for more quantitative microbiological applications and detection (Table 2). These more complex methods of detection often increase the cost of the test and limit its portability. Microbiology applications often employ benchtop analyzers that have microfluidic systems (Table 2) to enable sample preparation, mixing, heating with electrodes, or micro-electromechanical systems, as well as to facilitate DNA amplification and subsequent detection using fluorophore conjugation, chemiluminescence, or electrochemistry. A popular example of quantitative test methods is the droplet digital microfluidics platform [59,60] that uses fluid droplets to isolate and transport reagents and sample within the microfluidic cartridge. Processes like droplet sorting, coalescence, and mixing can all occur within a single cartridge. The complexity of the mechanical manipulation combined with the biochemistry involved may explain the higher operational failures observed in microfluidic devices (Figure 4). These failures are likely due to inconsistent performance of the device components and can have different implications depending on the microfluidic technology. For example, in digital droplet manipulation, high throughput droplet actuation, precise control of movement, coordinated rapid measurement, real-time analysis, and surface fouling all pose unique challenges [3,61,62]; whereas in systems with microfluidic pumps, particle accumulation and air bubble formation can be issues that result in inconsistent outcomes and operational failures. The reduced reaction volumes seen in microfluidics can also result in evaporation and sample stability issues. Alternatively, particle depositions inside microfluidic channels can lead to detrimental channel blockages and affect device efficiency, which might not be an issue when using non-microfluidic devices. Microfluidics-specific bench testing protocols to test the device performance over the entire range of operation can help to assess the potential occurrence of these issues. Introducing modularity in device components or developing test methods for the customized components [20] would help to evaluate these complex devices. Additionally, proper integration of different components and functions through a programmable, automated process is likely to help resolve operational issues [50]. Challenges in flow control and stability, improper mixing of reagents, evaporation during fluid transport, and dispersion are some flow-related issues [51] that could possibly be resolved through standard microfluidic test methods. 

Data output issues were observed for both quantitative and qualitative microbiological assays and COVID-19 diagnostic tests, accounting for more than 85% of all reported failures for qualitative, semi-quantitative, and quantitative tests. Microbiological devices come with pre-defined expectations around acceptable false positive and negative results, and erroneous results are always possible with laboratory tests [63]. Therefore, it is not surprising to see high occurrences of data output-related failure with EUAs. False negative results may be considered more problematic than false positive results since they are likely to cause a false sense of security and result in no treatment measures taken [64]; however, one of the ways to address these issues is by conducting serial or repeat testing [65]. Since virtually all the EUAs use small sample volumes and solely based on that metric can be classified under microfluidics, a technology specific failure mode analysis for microfluidics-based EUAs with non-microfluidic-based EUAs was moot and not performed. To establish a clear correlation between technology used and data output failures, further systematic comparisons across technologies and failures are needed. We can speculate that microfluidics, in theory, may exacerbate the number of data output failures. Non-specific absorption of proteins in microfluidic devices with low analyte volume makes the detection of biomarkers challenging resulting in false positives. Hydrophobic channel materials can accentuate this artifact. Alternatives to restrict non-specific absorption can result in limited interaction between the sample and biosensors in the microchannel and can cause false negative responses [3]. In POC or home use devices, these data output issues are likely to arise from several factors like transport, temperature, or humidity affecting the reagents or from inconsistent usability factors. Therefore, developing rigorous test methods for real-world use scenarios is likely to be beneficial for microfluidic medical devices [66].

Devices for diabetes management can be broadly divided into two types, for monitoring (e.g., continuous, and self-monitoring glucose systems) and for treatment (e.g., injection devices, insulin pumps, and software, such as automatic insulin dose calculators and controllers). Of these management options, only continuous glucose monitors and insulin pumps use micro-scale components and low flow rates. Continuous glucose monitors usually have tiny sensors that are inserted under the patient skin and sense glucose levels through chemical gradients and reactions. From a microfluidics standpoint, they are likely to be simple since they do not require manipulation of small volumes of liquids. The self-monitoring blood glucose monitors typically work by finger-stick, where the blood reaches the controller through capillary action. In this case, the blood can be drawn using microfluidic devices, such as microneedles, or through the use of polymer strips with microscale dimensions and is subsequently analyzed using devices that often have microchannels. These types of devices can be used at home much like continuous glucose monitors, and although they lack the ability to read continuously, they remain the standard of care for measuring glucose levels at home. The third type of diabetes management is the insulin pumps that deliver insulin. Insulin pumps are complex from a flow perspective in that they are likely to contain micro-pump systems that drive flow for delivering insulin using hydrodynamic pressure. As insulin pumps become more miniaturized and portable [67], the downside is that increasing pressure drop in the pump systems may increase the likelihood of leaking or mechanical integrity failures [11]. Indeed, most of the flow-related issues identified in Figure 4 for microfluidic devices are related to leakage. This is consistent with prior surveys that have identified the importance of flow related testing for microfluidic devices [9]. These issues can be resolved through leakage testing, for which standardized methods are available (ISO 28620:2010 and ISO 594-2). However, scaling down existing flow-related standardized methods may not always be appropriate [11,68]. The sensitivity required for ensuring pneumatic integrity and accurate flow measurement in microfluidic devices differs from that required for non-microfluidic devices. Microfluidic channels are connected to macro systems through connectors and this integration between macro and microsystems may result in leakage failures. Therefore, it is important to design test methods specific to microfluidic flow performance testing with higher sensitivity and small volume leakage detection. Efforts for developing standards for flow leakage and measurement are also mentioned in the recent publication by Cavaniol [69]. With proper calibration, outputs like pressure or flow rate data can be used as indicators for the system performance. For example, previous research [70] has indicated that monitoring fluctuations in pressure may be an effective tool for optimizing performance of a microfluidic system. Overall, standardization efforts for flow-related testing will be useful for the microfluidic device community.

Clotting, cell counting, and ion detection applications mostly use blood, and generally incorporate a standalone platform with microfluidic components where collection and mixing with reagents occurs and is then analyzed. POC devices use microfluidic cartridges with only a few microliters of sample that work in conjunction with analyzers. These types of devices almost always use either pneumatic-driven or pump-based flow. For these applications, there are no prominently used flow or sensing systems, although ion detection sensing is often accomplished through chemical gradients (Table 2). Manufacturers submitting devices for this application often provide test data to support the usability and operation of the device, with a focus on bubble detection [71], channel blockage, sample handling, transport, and sample loading—all of which are commonly reported issues associated with microfluidic technology. The use of blood as an analyte for diagnostic tests also makes hemocompatibility testing critical. Improper sample handling, large inter-device variability due to variations in design and coagulation reagents, and variations in device response due to variability in blood viscosity and hemolytic conditions result in operation-related and data output failures [72]. Therefore, defining standardized handling practices and proper tests to assess reproducible device performance are valuable for hematological devices.

### 4.2. Detection Methods, Materials and Samples

Independent of the application or the type of flow mechanism, the detection method used by a microfluidic device is an equally important consideration. As seen in Table 2, chromatography is widely used for detection due to its simplicity in operation but provides limited quantitation. Therefore, other detection technologies are required in microfluidics-based devices depending on the application. For electrical and electrochemical sensing, the durability of the micro-electrodes may need to be assessed, proper cleaning protocols to avoid fouling of the electric circuit should be used, and the detection sensitivity over time of use should be characterized [73]. It is also important to ensure that electric sensing is non-invasive and does not result in sample fouling or damage, as this may affect downstream analysis [62]. Electrokinetic phenomena are sensitive to buffer conductivity, which may result in large performance fluctuations for POC devices. Therefore, impedance measurements for testing the buffer conductivity are required to optimize the device performance [3]. For systems using optical components for detection, integration of the optical readout with microfluidic platforms is a major challenge. Since the micro-scale options for obtaining optical data are limited, non-linear methods with external equipment are often used which complicate data interpretation. Integration of external attachments for detection can cause mechanical vibrations, overheating, and electrical effects, thereby affecting system stability at the microscale. These systems, therefore, may need to be tested adequately after fabrication as well as during in-field operation [74].

Note that a device can often be made with different materials. For example, LFAs can use paper for harnessing capillary forces and could employ plastic as an external shield to protect the assay from external mechanical forces. To assess part-wise reliability of their devices, some manufacturers (Table 2) develop and perform individual tests for critical parts of the device (e.g., membranes, valves, bellows), ensuring that the performance of these components remains consistent over the entire device life cycle, as well across the entire operating ranges [75]. Eventually, developing standardized protocols may help reduce the time required for design specific protocols for various materials used. However, when considering such protocols, and given the diversity in the types of materials used for device fabrication, attributes like quality of fabrication, design tolerances, material compatibility, surface properties, wettability, and interconnections should be taken into consideration [9,76].

The microfluidics community struggles with a major disconnect between lab-based prototyping and final, mass-produced commercial products. Some materials, despite being cheap and easy to use, are not popular in the industry because of their limitations in scaling up production and undesirable material properties (e.g., PDMS). New materials choices take time to be adopted by the industry as the analytical and clinical validity, biocompatibility, material–fluid interactions, and reliability need to be thoroughly vetted. 

Whole blood, serum, plasma, urine, and nasopharyngeal swabs appear to be the most common fluids used by microfluidic medical devices. During device development, it may be acceptable to start by using a surrogate fluid that mimics the actual fluid, but, eventually, during the later stages of device development, it is often beneficial to use actual patient samples. In addition, to mimic the real-world use scenario, the impact of cold storage and transport of samples should be considered during any performance testing. Optimization of the sample preparation step may also be desirable by pretreating the analyte [77]. Inter-laboratory testing, testing at different POC sites, and simulation of different POC settings are also important considerations. For devices that use blood (e.g., blood coagulation devices), hemocompatibility studies may be needed to assess how different patient blood samples react under mechanical stress for particular device designs, materials, and surface coatings [78,79].

Several of the applications in microfluidics use only small sample and reagent volumes that range from a few µL to >100 µL (Table 2). For accurate clinical diagnosis, efficient transport of the analyte volume, proximity of the sample to the reagents for reducing the assay analysis time, and reproducible and precise delivery of reagents in a localized manner all may impact device performance [80]. Therefore, bench-top studies characterizing the transport of sample volumes and reagent release may help address reliability issues specific to small volume and rapid analysis applications. 

While some manufacturers choose to develop their own custom testing protocols for assessing flow performance (Table 2), developing test methods that use capillary, pneumatic and pump-based flows are likely to have a major impact on the field of microfluidic devices (Figure 3b). It is important to evaluate flow consistency, repeatability of the flow cycle, priming of the pump, and backpressure in the system. Additionally, challenges in flow control and stability, improper mixing of reagents, evaporation during fluid transport, and dispersion are some issues associated with the anticipated flow mechanisms [79].

### 4.3. Limitations

While we attempted to be thorough with our datamining by including information from medical devices submissions to the FDA, as well as public database and literature sources, our search results are likely to have omitted some microfluidics-based technologies. This potential limitation is due to a lack of a consensus on the list of common terminologies used for defining microfluidics-based devices, or even devices that have some microfluidic components. The need for harmonization of terminologies in this field has been long felt by the community, and resulted in ISO efforts [81], and more recent efforts by the organ on a chip community; however, widespread adoption of microfluidics specific standards in the community is still lacking [39].

Another limitation of our study involves the failure mode analysis. Note that the FDA MAUDE database is a passive surveillance system, and this system cannot be solely used to establish accurate failure rates for devices. Deriving a causal relationship and trends using MDRs should be performed with caution. Underreporting of adverse events through a passive surveillance system is commonplace and we further expect this trend to be exacerbated in the at home diagnostics space as consumers are more likely to simply repeat a failed test rather than take the steps to report a malfunction. Therefore, findings from our failure analysis should be interpreted with caution. We assumed uniform reporting practices by manufacturers, which may not be observed in practice. For example, some manufacturers may file reports only if patient death or serious harm occurs, while other manufacturers may file adverse events even for inconvenient malfunctions that do not impact device safety. We also assumed that better design verification will result in fewer on-market issues. There is the possibility that on-market failures may result from manufacturing reliability issues and may not occur due to problems that were overlooked during design verification (which may have used devices built using pilot-scale manufacturing systems). These issues can be better tackled with manufacturing controls.

Because of logistical reasons, the adverse event analysis for non-microfluidic devices was not collected and analyzed for the time period of October 2021 to December 2021. However, given the large number of events analyzed under this category (>1 million events), we assumed that the contribution of adverse events for the omitted time period would not significantly affect the normalized failure mode analysis provided in Figure 4. 

Several manufacturers did not provide sufficient information on device characteristics, such as the type of pumps their devices employed. Therefore, we were unable to fully categorize microfluidic devices based on some factors like the types of pumps that they use.

## 5. Summary and Future Outlook

Analyses for determining the landscape of microfluidic medical devices have been limited so far. In addition, an analysis of failure modes in commercial microfluidic devices had never been performed. We conducted the first systematic review of the medical device submission landscape in the U.S., with the objective of identifying the current technology trends and areas where test method development may be beneficial. 

Applications: For qualitative and quantitative assays, developing bench testing protocols specifically intended for LFAs and protocols for modular systems for use during device development phase and subsequent scale up may be beneficial. For other popular applications (e.g., cell counting, ion detection), standardized handling practices specific to the analyte of interest may be beneficial.Failure Modes: For complex flow based microfluidic devices, rather than leveraging existing standards developed for traditional technologies, developing specific protocols (e.g., leakage testing) would be advantageous. More rigorous comparisons are required to understand if the data output issues in microfluidics-based microbiology diagnostic devices are similar to traditional assays. If the devices using microfluidics are found to be more prone to usability related failures, developing protocols for these types of tests may be beneficial. Miscellaneous: Risk management analyses around failure modes can sometimes be deployed as a strategy to improve the reliability of microfluidics based medical devices. The impact of miniaturization on detection methods needs to be investigated thoroughly. Lastly, testing protocols should mimic real-world use scenarios, including the use of the relevant sample matrix and the material to be used in the final finished form of the commercial device. 

This analysis was performed to encourage discussion around the challenges and unique aspects in developing microfluidics based medical devices and bringing them to market. While the community has started to develop consensus standards in the field of microfluidic medical devices, much work remains to be completed, particularly around development of new test protocols. It is likely that some of the device manufacturers who now have authorized EUAs will submit 510(k) applications for their devices, so that they can continue to be legally marketed in the U.S. after the pandemic. Given that testing requirements for 510(k) devices are generally more rigorous compared to EUAs, it is timely to begin developing test protocols to effectively evaluate these devices. In addition, the challenges posed by COVID-19 showed that the medical device community is highly capable of developing useful devices that can quickly benefit patients. Therefore, the community should strive to accelerate the time from innovation to market for safe and effective microfluidic medical devices.

## Figures and Tables

**Figure 2 micromachines-14-01293-f002:**
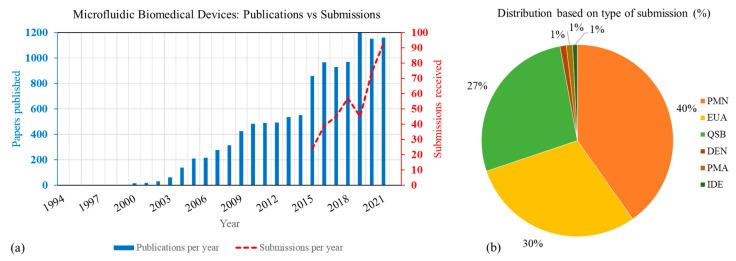
(**a**) Trend in number of publications for the keyword search ‘microfluidics + medical + devices’ and device submissions received at CDRH. (**b**) Type of medical device submissions containing microfluidics technology. PMN: Premarket Notification, EUA: Emergency Use Authorization, QSB: pre-submissions, DEN: De Novo Classification Request, PMA: Premarket Approval, IDE: Investigational Device Exemp-tion.

**Figure 3 micromachines-14-01293-f003:**
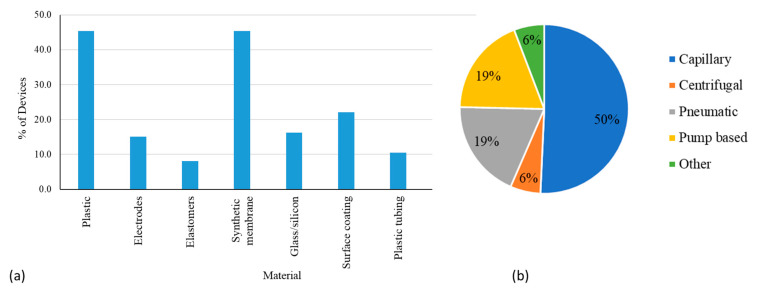
(**a**) Types of materials used in microfluidic medical devices. Note that one microfluidic device can be made up of multiple materials, and hence, the sum of the percentages in materials exceeds 100%. (**b**) Different flow mechanisms in microfluidic medical devices.

**Figure 4 micromachines-14-01293-f004:**
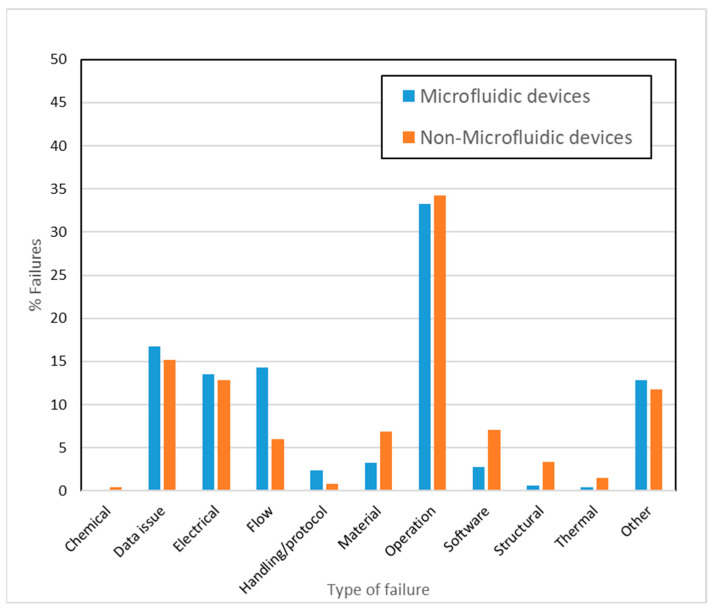
Percentage of types of failures in microfluidic and non-microfluidic devices during the period of 2015 to 2021 (excludes EUAs).

**Table 2 micromachines-14-01293-t002:** Prominent microfluidic applications, along with their corresponding device characteristics. The applications are organized in descending order based on the number of device submissions. POC: Point-of-care; OTC: over the counter; Rx: Prescription use. The product codes listed in Section 2.3 were used across multiple databases (e.g., [26,27,32,34]) to populate this table. For swabs the volume reported is after adding reagents.

Application	Fluid Types	Typical Sample Volume	Typical Assay Time/Duration of Use (min)	Location of Use	Fluid Transport Mechanism	Microscale Components	Methods of Detection	Types of Bench Testing conducted that also Assessed Microfluidic Functionality
COVID-19 Diagnostics	Blood, serum/plasma, nasal swab	3–5 drops	15–30	POC, Lab, Home (OTC)	Capillary, pneumatic	Coated nitrocellulose membrane used in lateral flow assays; microfluidic cartridge, and components for polymerase chain reaction; integrated fluid circuits	Chromatography; Chemiluminescence; Electrochemistry; Fluorescence	Hazard analysis, Transport and shelf-life testing including temperature/humidity effects Usability (Flex studies): sample volume variability, sample dilution, visibility of output
Coagulation/Prothrombin Time	Blood, plasma	<10–300 µL	<1 min	POC	Capillary, pneumatic, pumping	Microfluidic cartridge, microfluidic liquid handling,	Amperometry; Electrochemistry; Clotting, Chromogenic, Immunochemical	Quality checks that run during sample analysis; Shelf life and sample handling testing
Glucose Measurement	Blood	−1 µL	<1	Lab, POC, Home (OTC, Rx)	Capillary, electro-wetting, pneumatic	Micro-needle, lateral flow assays, microfluidic cartridge	Amperometry; Electrochemistry	Operation and flow testing: Sample volume study, Error detection features for filling problems Transport and shelf-life testing: temperature/humidity/altitude effects Usability: testing with used strips, drop, vibration
Respiratory Viral Assay and Multiplex systems	Nasal/throat swab, blood	50–500 µL	15–300	Lab, POC	Capillary, pneumatic, pumping	Microfluidic cartridge, multiport valves, elastic bladders	Polymerase chain reaction; Fluorescence; Optical detection	Reproducibility Error Monitoring or Internal quality controls Validation and Verification based on Risk Analysis
Bacterial and Fungal Assays	Nasal/throat swabs, urine, rectal-vaginal swab, blood	50–350	10–120	POC, Home (OTC, Rx)	Centrifugal, electro-wetting, pneumatic, pumping,	Microfluidic liquid handling, elastic bladders, automated pipetting	Polymerase chain reaction; Fluorescence; Voltametry; Optical detection	Operation and flow testing: reproducibility, automated sample loading, automated processing and purification
Infusion Pumps	Insulin	0.25–300 µL/h	3 days	Home (Rx)	Pumping	Functionalized microfluidic tubing, circuits for micro-dosing	Displacement	Operation and flow testing: priming, occlusion detection, delivery accuracy, mechanical integrity, stability, flow performance, bolus dosing, leakage Risk Management Transport and shelf life: temperature/humidity effects including rapid changes Usability: Simulated use
Pregnancy Test Kits	Serum/plasma, urine	25–80	≤10	POC, Home (OTC)	Capillary	Coated nitrocellulose membrane, fiber glass pad, microfluidic cartridge	Chromatography, chemiluminescence	Transport and shelf life: stability, simulated transportation Usability: sample volume adequacy, sample loading
Cell Counting	Blood	30–40	≤10	Lab, POC	Capillary, pneumatic	Microfluidic channel, microfluidic components for sample loading	Automated imaging inspection and analysis	Shelf life, shipping, stability testing, repeatability and reproducibility
Ion Detection	Blood, urine	1–3 drops	5–20	POC	Electro wetting, pneumatic	Microfluidic cartridge, micro- electrodes, microfluidic mixing chambers	Amperometry, Photometric, chemistry	Reproducibility, shelf life, transport, quality checks for mechanical components

**Table 3 micromachines-14-01293-t003:** List of commonly observed failures modes.

Type of Failure Mode	Description of the Issue
Chemical	Contamination of device constituents or reagents and presence of particulates
Data Output	Inconsistencies in data obtained from the devices such as low or high results, non-reproducible results, incorrect measurements, false positives, false negatives, and inadequate data
Electrical	Failures related to battery operation, circuit, loss of connections and power failure, problems related to electrical shorting, electric shocks, electric sparking, and electronic component failures
Flow	Leakage, splashing, occlusion, partial or total blockages, failures related to back flow, improper infusion, inaccurate flow rate, restricted flow, or excessive flow
Handling/Protocol	Failures occurring due to improper device handling or unclear instructions as per user, incorrect commands, and incorrect operating procedures from the user
Material/Structural	Devices issues related to breaking of components, fitting and assembly related issues, issues with improper bonding of different components and joint failures, which often happen because of failures related to bending and fracture of material, corrosion, material failures due to deformation, fragmentation, fraying, and disintegration
Operational	Operational failures include mechanical issues with device operation, pump failures, failure in alarm systems, inability to produce results, improper activation, issues with sensors, calibration, and compatibility problems
Software	Issues related to display, connection of device to hardware, interoperability failures, issues with device programming, software installation
Thermal	Failures related to humidity and temperature, issues related to heating of device or charring, melting of components due to over-heating
Other	Failures related to improper packaging, product quality problems and adverse events where no apparent cause is reported

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
