# Peer review of "A Systematic Analysis of Recent Technology Trends of Microfluidic Medical Devices in the United States"

_micromachines, 2023, doi:10.3390/mi14071293_

Round 1

Reviewer 1 Report

The paper by Natu et al. provided a systematic analysis of technology trends of microfluidic medical devices in the US, specially focusing on the commercial perspective. Lots of statistical data has been clearly presented in tables and charts. This paper is expected to identify the current technology trends and areas where test method development may be beneficial.

I suggest to add some discussion regarding the fabrication and integration. For example, there should be difficulties in balancing the versatility and ease of integration.

Author Response

The paper by Natu et al. provided a systematic analysis of technology trends of microfluidic medical devices in the US, specially focusing on the commercial perspective. Lots of statistical data has been clearly presented in tables and charts. This paper is expected to identify the current technology trends and areas where test method development may be beneficial.

Response: We sincerely appreciate the time and effort in providing review of this manuscript. We also appreciate the constructive feedback provided by reviewer 1.

I suggest to add some discussion regarding the fabrication and integration. For example, there should be difficulties in balancing the versatility and ease of integration.

Response: We have included the following changes to the discussion section of the manuscript in Section 4.2 Detection Methods, Materials and Samples. The document with tracked changes (yellow highlights) is attached with this response (page 18/28). 

The microfluidics community struggles with a major disconnect between lab-based prototyping and final, mass-produced commercial products. Some materials despite being cheap and easy to use, are not popular in the industry because of their limitations in scaling up production and undesirable material properties (e.g. PDMS). New materials choices take time to be adopted by the industry as the analytical and clinical validity, biocompatibility, material-fluid interactions, and reliability need to be thoroughly vetted.  

Reviewer 2 Report

The authors aimed to reveal the technology trends of microfluidic devices with the aid of data mining and analysis of firsthand information specifically toward biomedical use from a regulatory standpoint of the U.S. Food and Drug Administration. One of the ultimate goals of the development of microfluidic devices is to make them accessible to the general public, who may be outside the professional field, via commercializing integrated systems. There have been many outstanding review papers from academia and industry but few from the government regulatory agency. This well-written paper may offer many unique and interesting opinions for diverse readerships. I would like to endorse publication if the following issues can be properly addressed.

(1) A part of Figure 1 is missing in the manuscript for peer review.

(2) The title of Table 2 contained some missing text.

(3) PDMS is an elastomer. So the category axis labels of Figure 3a should be refined.

(4) As “trend” may refer to a recent, current, or future trend, this term should be clearly defined in the beginning in stead of in the last part of this paper. As found in this paper, the majority of the technology applied for the sample delivery is the old (but reliable) lateral flow, representing a current (or continued) trend. Conversely, saliva is always an alternative and convenient specimen for many POCTs but seemly still a minority among others, which may not simply mean that saliva has no potential as a new trend. Such ambiguity should be carefully clarified, allowing the readers and developers to understand the trend this paper is trying to reveal. 

(5) 300+ device submissions were included in the analysis. The total number was not so large. It would be helpful for readers if the full list could be accompanied by this paper as an appendix in a concise Excel file, allowing the readers to reach a specific submission of interest in the public database.

(6) A part of the legend in Figure 4 is missing.

Author Response

The authors aimed to reveal the technology trends of microfluidic devices with the aid of data mining and analysis of firsthand information specifically toward biomedical use from a regulatory standpoint of the U.S. Food and Drug Administration. One of the ultimate goals of the development of microfluidic devices is to make them accessible to the general public, who may be outside the professional field, via commercializing integrated systems. There have been many outstanding review papers from academia and industry but few from the government regulatory agency. This well-written paper may offer many unique and interesting opinions for diverse readerships. I would like to endorse publication if the following issues can be properly addressed.

Response: We appreciate the constructive feedback provided by reviewer 2, and we echo the opinion expressed. We sincerely appreciate the time and effort in providing a thorough review of this manuscript.

  • A part of Figure 1 is missing in the manuscript for peer review.

Response: We see that part of the Figure 1 got cut off when the editorial team formatted the manuscript. We sincerely apologize the inconvenience this may have caused and have now replaced the previous figure 1 with a more compressed figure for easier visualization of the reader and reviewer. Please refer to the updated Figure 1 in the attached document (manuscript with tracked changes). 

  • The title of Table 2 contained some missing text.

Response: We sincerely apologize for this oversight and have now updated the Table 2 title. Please refer to document attached. 

  • PDMS is an elastomer. So the category axis labels of Figure 3a should be refined.

Response: we agree with the reviewer and apologize for the oversight. We have included PDMS as part of Elastomers and replotted Figure 3a.

  • As “trend” may refer to a recent, current, or future trend, this term should be clearly defined in the beginning in stead of in the last part of this paper. As found in this paper, the majority of the technology applied for the sample delivery is the old (but reliable) lateral flow, representing a current (or continued) trend. Conversely, saliva is always an alternative and convenient specimen for many POCTs but seemly still a minority among others, which may not simply mean that saliva has no potential as a new trend. Such ambiguity should be carefully clarified, allowing the readers and developers to understand the trend this paper is trying to reveal. 

Response: We have made minor modifications in the title, abstract, and introduction by clarifying that we mean to represent “recent” trends. These are highlighted in yellow in the attached document. Note that our analysis (summary in Figure 1) includes device submissions over a 6-year period (2015-2021) with the objective to prepare for future regulatory science needs. We hope the changes made add clarity to the manuscript.  

(5) 300+ device submissions were included in the analysis. The total number was not so large. It would be helpful for readers if the full list could be accompanied by this paper as an appendix in a concise Excel file, allowing the readers to reach a specific submission of interest in the public database.

Response: The reviewer makes an excellent suggestion. However, as a regulatory agency that reviews confidential data this suggestion poses a challenge for us. In fact, after a thorough deliberation that included our senior management, we decided to not to pursue providing specifics and also not to include submissions that are not yet public information (e.g. presubmissions). While FDA strives for transparency, any mention or representation of a particular device in this paper or in a supplementary file may be interpreted as an endorsement of that device. Thus, to avoid any perceived biases towards a specific device or a device manufacturer, we won’t be able to make any changes in response to the comment. However, as the community moves towards defining terminologies in microfluidics, and more manufacturers adapt using this term in the publicly available portion of their submissions (510k summaries), we hope that such information can then be derived more easily from the existing public databases hosted by FDA 510(k) Premarket Notification (fda.gov). A reader who is interested in learning more about microfluidic medical device can do so by using the various product codes listed in section 2.3 and looking them up in the publicly available databases on FDA’s website (as suggested in Table 2’s updated title).

(6) A part of the legend in Figure 4 is missing.

Response: Thank you for identifying this error. We have updated Figure 4 with the missing legend. Please refer to attached document. 
